# Identification and Isolation of α-Glucosidase Inhibitors from *Siraitia grosvenorii* Roots Using Bio-Affinity Ultrafiltration and Comprehensive Chromatography

**DOI:** 10.3390/ijms241210178

**Published:** 2023-06-15

**Authors:** Fenglai Lu, Jiayi Sun, Xiaohua Jiang, Jingru Song, Xiaojie Yan, Qinghu Teng, Dianpeng Li

**Affiliations:** 1Guangxi Key Laboratory of Plant Functional Phytochemicals and Sustainable Utilization, Guangxi Institute of Botany, Guangxi Zhuang Autonomous Region and Chinese Academy of Sciences, Guilin 541006, China; lufenglai@gxib.cn (F.L.); sunjy0212@163.com (J.S.);; 2Guangxi Key Laboratory of Electrochemical and Magnetochemical Functional Materials, College of Chemistry and Bioengineering, Guilin University of Technology, Guilin 541004, China

**Keywords:** *Siraitia grosvenorii* roots, α-glucosidase inhibitor, ultrafiltration, chemical constituents, molecular docking

## Abstract

The discovery of bioactive compounds from medicinal plants has played a crucial role in drug discovery. In this study, a simple and efficient method utilizing affinity-based ultrafiltration (UF) coupled with high-performance liquid chromatography (HPLC) was developed for the rapid screening and targeted separation of α-glucosidase inhibitors from *Siraitia grosvenorii* roots. First, an active fraction of *S. grosvenorii* roots (SGR2) was prepared, and 17 potential α-glucosidase inhibitors were identified based on UF-HPLC analysis. Second, guided by UF-HPLC, a combination of MCI gel CHP-20P column chromatography, high-speed counter-current countercurrent chromatography, and preparative HPLC were conducted to isolate the compounds producing active peaks. Sixteen compounds were successfully isolated from SGR2, including two lignans and fourteen cucurbitane-type triterpenoids. The structures of the novel compounds (**4**, **6**, **7**, **8**, **9**, and **11**) were elucidated using spectroscopic methods, including one- and two-dimensional nuclear magnetic resonance spectroscopy and high-resolution electrospray ionization mass spectrometry. Finally, the α-glucosidase inhibitory activities of the isolated compounds were verified via enzyme inhibition assays and molecular docking analysis, all of which were found to exhibit certain inhibitory activity. Compound **14** exhibited the strongest inhibitory activity, with an IC_50_ value of 430.13 ± 13.33 μM, which was superior to that of acarbose (1332.50 ± 58.53 μM). The relationships between the structures of the compounds and their inhibitory activities were also investigated. Molecular docking showed that the highly active inhibitors interacted with α-glucosidase through hydrogen bonds and hydrophobic interactions. Our results demonstrate the beneficial effects of *S. grosvenorii* roots and their constituents on α-glucosidase inhibition.

## 1. Introduction

Diabetes mellitus (DM), a metabolic disease characterized by sustained hyperglycemia resulting in various complications, has gradually become one of the most serious metabolic disorders worldwide [1]. Inhibiting carbohydrate digestive enzymes to delay glucose absorption is one of the most effective approaches for overcoming postprandial hyperglycemia. Therefore, α-glucosidase, a carbohydrate digestive enzyme in the small intestine that catalyzes the cleavage of oligosaccharides to glucose in the small intestine, is an essential target for the regulation of postprandial serum glucose in patients with diabetes [2]. To date, α-glucosidase inhibitors, such as acarbose, miglitol, and voglibose, have been recommended as first-line therapies [3]. However, these drugs may cause adverse reactions, such as nausea, vomiting, diarrhea, and damage to liver and kidney function [4,5]. Hence, the extraction of effective α-glucosidase inhibitors with low toxicity from medicinal and homologous plants has attracted increasing attention because of their wide range of sources, minimal side effects, and excellent health-promoting activities [6,7,8].

Cucurbitaceae is a well-known plant family, considered to have potent hypoglycemic effects, and is recognized in the empirical control of DM [9]. As a member of the Cucurbitaceae family, *Siraitia grosvenorii* is a precious medicinal and edible plant distributed mainly in southern China. *S. grosvenorii* fruits are used as traditional herbal medicines for the treatment of dry cough, extreme thirst, sore throat, and hyperglycemia. Several cucurbitane glycosides have been identified from the fruits and have been reported to have broad bioactivities, including antioxidant, anti-inflammatory, hepatoprotective, anti-tumor, and anti-asthma activities [10,11,12,13,14]. Moreover, cucurbitane glycosides can reduce blood glucose by increasing insulin levels and inhibiting α-glucosidase activity, thereby reducing pancreatic injury [15]. Although several *S. grosvenorii* roots are available, they have only been used as organic materials and fertilizers in plantations. A few phytochemical investigations have revealed that the ethanolic extract of the roots mainly consists of cucurbitane-type compounds, including cucurbitacin B, cucurbitacin Q1, siraitic acids A–H, siraitic acid II B, siraitic acid II C, and siraitic glycoside II F [16,17,18,19,20]. Our preliminary study revealed that *S. grosvenorii* root extract inhibits α-glucosidase, but the active constituents that inhibit α-glucosidase have not been determined yet. Therefore, the effective components and the mechanism involved in the hypoglycemic effects of *S. grosvenorii* roots need to be explored further.

An efficient isolation strategy plays a key role in active substance discovery. However, owing to the complexity of natural extracts and the high structural diversity of their components, the conventional separation strategy presents randomness and blindness. In recent years, affinity-based ultrafiltration (UF) coupled with high-performance liquid chromatography (HPLC) has emerged as a simple, rapid, and effective technique to identify bioactive candidate molecules from natural products. UF-HPLC inhibition profiling provides a biochromatogram that allows the identification of HPLC peaks correlated with inhibitory activity, and thereby, specific isolation toward only active peaks becomes possible. This technology has been successfully used to accelerate the identification of α-glucosidase, α-amylase, and aldose reductase inhibitors [21,22,23].

In this study, a UF-HPLC-based strategy was established to screen α-glucosidase inhibitors from *S. grosvenorii* roots, and the targeted separation of these potential inhibitors was guided via UF-HPLC and was performed using a combination of MCI gel CHP-20P column chromatography, high-speed countercurrent chromatography (HSCCC), and preparative HPLC (pre-HPLC). Furthermore, the inhibitory activities of the potential inhibitors were verified, and the underlying plausible mechanisms were further explored using molecular docking analysis. This study supports the potential application of *S. grosvenorii* roots in the prevention and treatment of diabetes.

## 2. Results and Discussion

### 2.1. Inhibition of α-Glucosidase by Crude S. grosvenorii Root Extract and the Eluted Fractions

Generally, the efficacy of medicinal plants depends on their characteristic bioactive chemical components. Since we had previously found that *S. grosvenorii* roots extract inhibited α-glucosidase, we obtained more active fractions using the HPD-100 macroporous resin column. In the present study, according to HPLC analysis, Fraction 1 and 2 (SGR1 and SGR2) of *S. grosvenorii* roots extract demonstrated significantly different constituents, which were eluted from the HPD-100 column with 40% and 60% MeOH, respectively. Moreover, an α-glucosidase inhibitory assay was carried out to assess the effect of SGR1 and SGR2, while acarbose served as a positive control. The assay results (Table 1) showed that SGR2 was more active than the crude extract and SGR1, with an IC_50_ value of 631.1 ± 50.2 μg/mL, which is superior to that of acarbose (860.3 ± 31.3 μg/mL). It has been reported that the main components in *S. grosvenorii* roots are triterpenoids, which could be enriched with a 60% methanol solution. This may explain why SGR2 exhibited the strongest α-glucosidase inhibitory activity. Consequently, there is an urgent need to rapidly identify natural α-glucosidase inhibitors from SGR2.

### 2.2. Screening for Potential α-Glucosidase Inhibitors in SGR2

Affinity UF is a new, highly sensitive, and throughput technique that can rapidly identify bioactive compounds from complex natural products [24,25]. After incubation with α-glucosidase and interception by a UF membrane, active compounds in SGR2 bound to α-glucosidae were released by adding methanol and then further analyzed using HPLC. Figure 1 shows that the 17 peaks in the SGR2 sample incubated with active α-glucosidase (Figure 1 blue line) showed higher intensities than the sample with denatured α-glucosidase (Figure 1 red line), implying that the active compounds in SGR2 are potential α-glucosidase inhibitors.

To further evaluate the relative binding capacities of individual compounds and their contribution to the overall inhibitory activity of the extract, the specific binding factor was calculated using Equation (2) [26]. The values of 17 specific binding factors obtained from the above calculation are shown in Table 2, which revealed that the 17 peaks exhibited significantly different binding capacities. A total of 6 peaks (9–12, 14–15) had specific binding factors higher than 5%, and peak 14 exhibited the highest binding affinity with a specific binding factor of 17.26%, followed by peak 15 with a specific binding factor of 7.98%. However, the detection of relative binding affinities did not completely represent the α-glucosidase inhibitory activities of the compounds. Thus, the α-glucosidase inhibitory activities of the identified ligands need to be further validated in vitro.

### 2.3. Isolation and Structural Identification

Isolation is an indispensable step in verifying the function of the active compounds. As shown in Figure 1, there were complex components with a broad range of polarities in SGR2, and some of them were in low abundance, which made it difficult to comprehensively isolate bioactive components via a single-step separation process. Therefore, guided by the UF-HPLC profiles, a combination of MCI gel CHP-20P column chromatography (CC), HSCCC, and pre-HPLC was conducted to isolate the active peaks associated with the α-glucosidase activity. First, to eliminate the untargeted constituents and concentrate the minor potential ligands, SGR2 was further fractionated using MCI gel CHP-20P CC. Macroporous resin CC is one of the most widely used techniques for pretreating crude extracts because of its high selectivity, efficiency, and simplified operation [27]. The eluate of the MCI gel CHP-20P CC was composed of five subfractions according to the HPLC analysis results, which were subsequently subjected to HSCCC and pre-HPLC processes. Pre-HPLC is an effective method for separating complex constituents with a broad range of polarities. In addition, it can provide a straightforward view of a specific analysis, which facilitates the precise isolation of compounds. Our results showed that pre-HPLC provided a high-resolution separation of peaks 4–12 from Fr 2.2–Fr 2.4, and peak 14 from Fr 2.5 (Figure 2). The peaks 1–3, 13, and 15–17 were distributed over a broad range in the HPLC chromatogram, but they all showed low and medium polarity, as observed by TLC analysis; therefore, they were distributed in the organic phase when partitioned by the aqueous phase. Previous studies show that HSCCC is highly effective in separating such compounds [17,19]. Hence, HSCCC was employed to separate peaks 1–3, 13, and 15–17. In summary, 16 compounds were isolated from SGR2 using comprehensive isolation techniques. However, the structure of peak 10 could not be elucidated because it was unstable after purification.

#### 2.3.1. Elucidating the Structure of Novel Compounds

Compound **4** was obtained as a white amorphous powder. The molecular formula was determined to be C_46_H_70_O_20_, based on HR-ESI-MS at *m*/*z* 987.4471 [M+HCOO]^−^ (calculated for C_47_H_71_O_22_, 987.4442). Its ^1^H NMR data (Table 3) in *pyridine-d*_5_ indicated the presence of five methyl groups [*δ*_H_ 1.95 (3H, s), 1.93 (3H, s), 1.53 (3H, s), 0.87 (3H, d, J = 6.2 Hz), and 0.76 (3H, s)], an olefinic proton at *δ*_H_ 7.38 (1H, t, J = 7.7 Hz), and three *β*-glucopyranosyl moieties [*δ*_H_ 6.39 (1H, d, J = 7.8 Hz), 4.99 (1H, d, J = 8.1 Hz), and 4.87 (1H, d, J = 7.8 Hz)] in compound **4**. Based on the ^13^C NMR data (Table 4), compound **4** contains 46 carbon atoms that construct three glucopyranosyl moieties (*δ*_C_ 105.5, 102.8, 96.5,78.9, 78.9, 78.7, 78.7, 78.6, 78.2, 75.4, 74.4, 73.5, 72.6, 71.7, 69.8, 64.0, and 62.8), three carbonyl carbons (*δ*_C_ 210.9, 198.9, and 167.6), one double bond (*δ*_C_ 146.3 and 127.2), five methyl groups (*δ*_C_ 19.2, 17.8, 17.6, 13.0, and 11.5), an oxygenated methine (*δ*_C_ 82.9), eleven methylene carbons (*δ*_C_ 69.8, 64.0, 62.8, 52.5, 40.0, 37.6, 33.8, 31.2, 28.8, 27.9, and 26.2) and four quaternary carbons (*δ*_C_ 158.2, 130.3, 51.1, and 47.7). These NMR signals indicate that compound **4** might be triterpenoid *β*-glucopyranoside. By comparing its ^13^C NMR data with those of siraitic acid E [20], three more signs for glycosylation and the glucosylation shifts of C-16 (*δ*_C_ 82.9) and C-27 (*δ*_C_ 167.6) were observed, suggesting that three sugar moieties might be attached to C-16 and C-27. In the heteronuclear multiple bond coherence (HMBC) spectrum, cross-peaks between H-1 (*δ*_H_ 4.87) of glucosyl-G_I_ and C-16 (*δ*_C_ 82.9) of the aglycone, and H-1 (*δ*_H_ 6.39) of glucosyl-G_II_ and C-27 (*δ*_C_ 167.6) were observed. Moreover, the correlation between H-6 (*δ*_H_ 4.73, 4.34, m) of glucosyl-G_II_ and C-1 (*δ*_C_ 105.5) of glucosyl-G_III_, as well as H-1 (*δ*_H_ 4.99) between glucosyl-G_III_ and C-6 (*δ*_C_ 69.8) of glucosyl-G_II_ confirmed the linkage of the G_III_-(1→6)-G_II_ moiety. Based on the above evidence, together with the analysis of the ^1^H-^1^H correlation spectroscopy (COSY) and nuclear overhauser effect spectroscopy (NOESY) spectra, compound **4** was determined as shown in Figure 3 and named siraitic acid III E. (The spectrograms of compounds **4** are shown in Appendix A).

Compound **6** was obtained as a white amorphous powder, which possessed a molecular formula of C_40_H_60_O_15_ deduced from HR-ESI-MS at *m*/*z* 803.3812 [M+Na]^+^ (calculated for C_40_H_60_O_15_Na, 803.3824). Fragment ions at *m*/*z* 618 [M-Na-Glc]^−^ and 456 [M-Na-2Glc]^−^ were obtained from the analysis of MS/MS fragment ions, suggesting the presence of two glucose moieties. The ^13^C and ^1^H NMR data for compound **6** (Table 3 and Table 4) were similar to those for compound **4**, except for the sugar moieties. The ^1^H NMR spectrum of compound **6** displays signals for two *β*-glucopyranosyl moieties [*δ*_H_ 6.50 (1H, d, J = 7.6 Hz), 4.88 (1H, d, J = 7.7 Hz)]. In the HMBC spectrum, the cross-peaks between H-16 (*δ*_H_ 4.52, t, J = 6.6 Hz) and C-1 (*δ*_C_ 102.8) of glucosyl-G_I_, H-1 (*δ*_H_ 4.86) of glucosyl-G_I_ and C-16 (*δ*_C_ 82.9) of the aglycone, and H-1 (*δ*_H_ 6.50) of glucosyl-G_II_ and C-27 (*δ*_C_ 167.6) of the aglycone suggested sugar linkage. Based on the abovementioned data, together with the analysis of the heteronuclear single quantum coherence spectroscopy (HSQC), HMBC, ^1^H-^1^H COSY, and NOESY spectra, the chemical structure of compound **6** was elucidated as shown in Figure 3 and named siraitic acid IIb E. (The spectrograms of compounds **6** are shown in Appendix A).

Compound **7** was obtained as white amorphous powder and yielded a quasi-molecular ion peak at *m*/*z* 803.3774 [M+Na]^+^ (calculated for C_40_H_60_O_15_Na, 803.3824) by HR-ESI-MS, consistent with a molecular formula of C_40_H_60_O_15_. The ^1^H and ^13^C NMR data for compound **7** (Table 3 and Table 4) were similar to those for compound **6**. The noticeable difference was that the signals of C-16 in **7** were downshifted from *δ*_C_ 77.0 to 82.8, indicating that compound **7** may be a C-16 isomer of compound **6**. The HMBC correlations between H-1 (*δ*_H_ 6.42) of glucosyl-GI and C-27 (*δ*_C_ 167.5) of the aglycone, H-6 (*δ*_H_ 4.37, 4.47, m) of glucosyl-G_I_ and C-1 (*δ*_C_ 105.8) of glucosyl-G_II_, and H-1 (*δ*_H_ 4.77) of glucosyl-G_II_ and C-6 (*δ*_C_ 69.8) of glucosyl-G_I_ constituted the G_II_-(1→6)-G_I_ moiety. In the NOESY spectrum, nuclear overhauser effect-related peaks of H-16 (*δ*_H_ 4.30) and H-20 (*δ*_H_ 1.61) were observed, and the relative configuration of the hydroxyl group was determined. Based on the abovementioned information and prior data, compound **7** is shown in Figure 3 and named siraitic acid II E. (The spectrograms of compounds **7** are shown in Appendix A).

The molecular formula of compound **8** was assigned as C_53_H_84_O_24_ by HR-ESI-MS at *m*/*z* 1103.5267 [M−H]^−^ (calculated for C_53_H_83_O_24_, 1103.5280). The NMR data (Table 3 and Table 4) showed signals attributable to six methyl groups (*δ*_H_ 1.91, 1.52, 1.11, 0.90, 0.82, 0.70, *δ*_C_ 27.3, 25.6, 18.8, 17.2, 17.1, and 13.0), two olefinic protons at (*δ*_H_ 7.06 and 5.51), together with four sugar anomeric signals *δ*_H_ (6.43, 5.15, 5.02, and 4.81) and *δ*_C_ (107.5, 106.0, 105.8, and 96.7). Compound **8** was classified as a triterpenoid saponin with a nor-cucurbitane skeleton according to previous studies [28,29], and its aglycone structure was the same as that of siraitic acid H. HMBC correlations between H-1 (*δ*_H_ 4.81) of glucosyl-G_I_ and C-3 (*δ*_C_ 88.2) of the aglycone, and H-1 (*δ*_H_ 5.16) of glucosyl-G_II_ and C-6 (*δ*_C_ 70.9) of glucosyl-G_I_ were observed, which constituted the G_II_-(1→6)-G_I_ moiety. Moreover, the cross-peaks between H-1 (*δ*_H_ 6.44) of glucosyl-G_III_ and C-27 (*δ*_C_ 167.6) and H-1 (*δ*_H_ 5.02) of glucosyl-G_IV_ and C-6 (*δ*_C_ 63.0) of glucosyl-G_IV_ constituted the G_IV_-(1→6)-G_III_ moiety. The NOESY spectrum showed that H-3 (*δ*_H_ 3.73) was related to H-9 (*δ*_H_ 2.18), H-9 was related to H-10 (*δ*_H_ 3.14), and H-9 was not associated with H-8. Therefore, the substitution configuration of the C-3 glycosyl group was determined. The relevant hydrocarbon spectrum data were analyzed and assigned using HSQC, HMBC, and ^1^H-^1^H COSY spectra. Subsequently, the structural formula of compound **8** was identified as shown in Figure 3, and named siraitic acid IV H. (The spectrograms of compounds **8** are shown in Appendix A).

Compound **9** was isolated as white amorphous powder, which gave a quasi-molecular ion peak at *m*/*z* 823.4124 [M+HCOO]^−^ (calculated for C_42_H_63_O_16_, 823.4122) by HR-ESI-MS, consistent with a molecular formula of C_41_H_62_O_14_. The ^1^H and ^13^C NMR data (Table 3 and Table 4) for compound **9** were similar to those for siraitic acid II C (compound **14**) [16], except for one additional methyl group in compound **9** (*δ*_C_ 23.2, *δ*_H_ 0.76 (3H, s)). Furthermore, this methyl signal (*δ*_C_ 23.2) was strongly correlated with H-8 (*δ*_H_ 1.90), C-9 (*δ*_C_ 50.0), C-10 (*δ*_C_ 41.1), and C-11(*δ*_C_ 214.8); C-9 was a quaternary carbon, thereby the methyl was attached to the C-9 position of the aglycone, which was determined to be the same as that in siraitic acid G. Therefore, the chemical structure of compound **9** was elucidated as shown in Figure 3 and named siraitic acid II G. (The spectrograms of compounds **9** are shown in Appendix A).

Compound **11** was obtained as white amorphous powder and had a molecular formula of C_41_H_64_O_15_ deduced from the quasi-molecular ion peak at *m*/*z* 819.4166 [M+Na]^+^ (calculated for C_41_H_64_O_15_Na, 819.4137) by HR-ESI-MS. HMBC examination revealed the occurrence of an epoxy functional group between C-19 and C-5 and a carbonyl group at C-11, which is consistent with the parent nucleus of siraitic acid A [30]. Compound **11** was similar to siraitic acid II B (compound **12**) [16], except that a hydroxyl group was placed at the C-3 position. Furthermore, it was observed that the carbon signal at C-3, in comparison with siraitic acid II B, evidently shifted to *δ*_C_ 78.3 in the ^13^C NMR data for compound **11**. Thus, based on comprehensive ^1^H-^1^H COSY and HMBC correlation analysis, compound **11** was determined as shown in Figure 3 and named siraitic acid II A. (The spectrograms of compounds **11** are shown in Appendix A).

The key ^1^H-^1^H COSY (red bold lines) and HMBC (blue arrows) correlations of the novel compounds are shown in Figure 4.

#### 2.3.2. Identification of the Known Isolated Compounds

The other 10 known compounds were identified to be (-)-lariciresinol (**1**) [31], 3,4′-dimethoxy-4,9,9′-trihydroxy-benzofuranolignan-7′-ene (**2**) [32], 23,24-dihydrocucurbitacin F (**3**) [33], siraitic glycoside II F (**5**) [17], siraitic acid II B (**12**) [16], 23,24-dihydrocucurbitacin F-25-acetate (**13**) [34], siraitic acid II C (**14**) [16], cucurbitacin B (**15**) [35], 23,24-dihydrocucurbitacin B (**16**) [36], and dihydroisocucurbitacin B-25-acetate (**17**) [37], respectively, based on their ^1^H and ^13^C NMR spectroscopic data (Appendix A), as well as the comparison of these data with those reported previously.

### 2.4. Inhibition of α-Glucosidase and Structural-Activity Relationship

Although the UF-HPLC assay was selective and specific for screening α-glucosidase ligands, false positives might arise from the nonspecific binding of certain compounds to nonfunctional α-glucosidase sites. To verify the potency of the abovementioned inhibitors, all isolated compounds were tested for α-glucosidase inhibition in vitro, at 1.0 mg/mL concentration (Table 2). Expectedly, all compounds exhibited different inhibitory actions. Compounds that displayed more than 50% inhibition were evaluated for IC_50_. Compounds **1**, **2**, **7**, **9**, **11**, **12**, **14**, and **15** demonstrated significant in vitro α-glucosidase inhibitory activity, with IC_50_ values in the range of 430.13–2275.47 μM. Among these, compound **14** (siraitic acid II C) exhibited the strongest α-glucosidase inhibitory activity. This study is the first to report their inhibitory effects against α-glucosidase.

It is known that the structural diversity of active molecules leads to significant differences in α-glucosidase inhibitory activity. The isolated compounds included lignans, cucurbitacins, and cucurbitane glycosides. A limited structural-activity relationship was developed for all three classes of compounds. Compared with cucurbitane glycosides, lignans and cucurbitacins contain more hydroxy groups, which are distributed at different sites. Studies have verified that the number and sites of hydroxy groups in the molecular structures play very important roles in the inhibitory activity against α-glucosidase. Cucurbitacins, which are structurally diverse triterpenes found in members of the Cucurbitaceae and several other plant families, possess immense pharmacological potential [38]. Cucurbitacin B was reported to exhibit anticancer and antidiabetic activities [39] and showed better inhibitory activity against α-glucosidase than the other cucurbitacins in this study.

Among the cucurbitane glycoside derivatives, compounds with two glucose residues showed better activity than those with three or four glucose residues. Zhang et al. revealed that deglycosylation could occur in cucurbitane glycosides (mogrosides) of *S. grosvenorii* fruits in T2DM rats and confirmed that the hypoglycemic effects of the mogroside fraction containing 1–3 glucose residues were superior to that containing 4–5 glucose residues [15]. Relevant studies have also found that flavonoid deglycosylation increases the inhibitory activity against α-glucosidase [23,40]. This may be due to the larger size of the glycosylated side chains and greater steric hindrance, resulting in a significant decrease in the binding affinity toward α-glucosidase. However, the activity difference between the compounds with two glucose residues (siraitic glycoside II F, siraitic acid IIb E, siraitic acid II E, siraitic acid II A, siraitic acid II B, and siraitic acid II C) may be due to the difference in the site of hydroxy groups and their steric conformation.

### 2.5. Molecular Docking Analysis

Molecular docking is a significant method to predict and study the interactions between ligand-receptor complexes by studying the docking energy, acting sites, and key residues [41]. The underlying mechanisms of the screened inhibitors (compounds **1**, **2**, **7**, **9**, **11**, **12**, **14**, and **15**) were further clarified by molecular docking analysis, the results of which are shown in Figure 5 and Table 5. Figure 5A–I shows the 3D structures of the docking of the eight inhibitors/acarbose with α-glucosidase. Table 5 lists the molecular docking results and shows that compounds (**1**, **2**, **7**, **9**, **11**, **12**, **14**, and **15**) possessed a lower or equal binding affinity to acarbose, which indicated that these active compounds bound closely to the receptor for α-glucosidase. However, except for compounds **1** and **14**, the inhibitory activities of the other compounds were lower than that of acarbose, despite their high binding energy values. This can be mainly attributed to the standard error of the Autodock program (2.5 kcal/mol) [42]. Notably, compound **14** demonstrated the strongest affinity to α-glucosidase (−10.0 kcal/mol), confirming that it was the strongest α-glucosidase inhibitor among the compounds isolated in the present study.

The docking results showed that all active compounds were well accommodated in the active site of α-glucosidase, through hydrogen bonding and/or hydrophobic interactions. With respect to specific docking sites, acarbose showed nine H-bond interactions with seven amino acid catalytic residues of α-glucosidase, including Gln 279, Asp352, Arg 442, Glu 411, Pro312, His 280, and Lys 156. Many studies have verified that Gln 279, Asp352, Arg442, Glu411, and His 280 are essential active catalytic sites of α-glucosidase [23]. The eight components formed multiple H-bonds with different amino acid residues of α-glucosidase. However, there were four common active residues (Asp 352, Arg 442, Gln 279, and His 280) that interacted with acarbose and the active isolated components, validating the established method.

In addition to H-bond interactions, hydrophobic effects also contributed to the interactions between α-glucosidase and the isolated compounds. However, only H-bond interactions were observed between acarbose and α-glucosidase, indicating that the isolated compounds possessed different active mechanisms against α-glucosidase compared with acarbose. The structural features observed in the isolated compounds included the presence of an electron-donating group, such as hydroxyl, carbonyl, and methyl groups, which bonded with the amino acid residues of α-glucosidase. For example, compound **14**, which was the most active compound, formed seven H-bonds with four catalytic residues, Gly 309, Arg 315, Asp 352, and Arg 442, of α-glucosidase, with an average H-bond distance of 3.199 Å (Figure 5G and Table 5). On the one hand, compared with compound **14** (IC_50_ = 430.13 ± 13.33 μM), the lesser activity by compound **9** (IC_50_ = 1580.96 ± 12.54 μM) might be due to the introduction of a methyl group at C-9. This substitution undoubtedly increases steric hindrance, thereby inducing conformational changes and/or reducing accessibility to the active binding site [43]. On the other hand, although compound **7** formed more H- bond interactions than compounds **9** and **14**, it still exhibited relatively low α-glucosidase inhibitory activity, which may be due to the difference in the catalytic active sites interacting with α-glucosidase. However, it is unclear which active site residues inhibit the catalytic activity of α-glucosidase. In summary, it was verified that the hydrogen bonds and hydrophobic effects between the isolated compounds and α-glucosidase play important roles in the high inhibition of α-glucosidase. These interaction modes clearly indicate the inhibitory capability of all isolated compounds against α-glucosidase. The results of the molecular docking study further confirmed the reliability of the UF screening technique.

## 3. Materials and Methods

### 3.1. Materials and Reagents

*S. grosvenorii* roots were collected from Longjiang Township (Guilin, China) and identified by Fenglai Lu, an associate researcher at the Guangxi Institute of Botany. Acarbose (RFS-A01411804026, ≥98%) and p-nitrophenyl-α-D-glucopyranoside (p-NPG, K17A10B82914, S10137, ≥99%) were acquired from Chengdu Herb Purify Co., Ltd. (Chengdu, China) and Yuanye Biological Technology Co., Ltd. (Shanghai, China), respectively. We obtained α-glucosidase (#00001076200, G5003-100UN, 24.19 U/mg) from Sigma-Aldrich (Shanghai, China). Phosphate-buffered saline (PBS, No. 20211029, pH 7.2–7.4) was purchased from Solarbio Science & Technology Co., Ltd. (Beijing, China). Acetonitrile (HPLC grade) was obtained from Fisher Scientific (Waltham, MA, USA), while dichloromethane, methanol, ethanol, and other organic solvents and chemicals (analytical grade) were acquired from Xilong Scientific (Guangzhou, China). Ultrapure water was used for all experiments.

### 3.2. Extraction and Preparation of Fractions from S. grosvenorii Root Extract

*S. grosvenorii* root extracts (SGR) were prepared as described previously [17]. Briefly, air-dried and powdered roots of *S. grosvenorii* (10 kg) were immersed in 70% (*v*/*v*) ethanol, extracted three times every two weeks at 25 °C (2 × 50 L), and concentrated under reduced pressure. The extract (1 kg) was inserted into a macroporous resin HPD-100 column (2 kg, 55 × 10 cm i.d.) and eluted with gradient mixtures of MeOH and H_2_O (40:60, 60:40, and 100:00 *v*/*v*; each 2.5 L). Finally, the 40% and 60% MeOH fractions were condensed, and the isolates were named SGR1 and SGR2, respectively, for chemical analysis and biological activity research.

### 3.3. Assessment of α-Glucosidase Inhibition

The α-glucosidase inhibitory activity of the *S. grosvenorii* root extracts was determined using a method mentioned previously [44], with some modifications to it. Briefly, the sample was dissolved in PBS containing 10% dimethyl sulfoxide. A mixture of the tested sample (50 μL, different concentrations) and α-glucosidase (20 μL, 0.2 U/mL) was placed in a 96-well microplate and incubated at 37 °C for 5 min. Next, 20 μL pNPG (1 mM) was added, and the mixture was further reacted for 30 min at 37 °C and terminated through the addition of NaCO_3_ solution (50 μL, 0.2 mM). The percentage inhibition of α-glucosidase was determined by measuring the absorbance of the mixture at 405 nm and calculated by Equation (1):Inhibition (%) = [1 − (A_inhibitor_ − A_blank inhibitor_)/(A_control_ − A_control blank_)] × 100(1)
where A_inhibitor_, A_blank inhibitor_, A_control_, A_control blank_ are the absorbance values of the potential inhibitor and α-glucosidase, only the potential inhibitor without α-glucosidase, only α-glucosidase, and PBS buffer, respectively.

### 3.4. Screening and Identification of α-Glucosidase Inhibitors

Affinity UF was performed as previously described [45], with slight modifications. Briefly, a PBS solution of SGR2 (10 mg/mL, 800 μL) was mixed with α-glucosidase (1 U/mL, 400 μL) and incubated at 37 °C for 45 min. Next, the mixture was centrifuged at 10,000 rpm for 10 min using a UF membrane with a 10 kDa molecular weight cutoff, to collect the active component and α-glucosidase complexes. The complexes were washed 3 times with 600 μL of PBS (0.1 M, pH 7.2) to remove unbound compounds. A solution of 70% methanol (200 μL) was added to the residue and incubated at 4 °C for 20 min to release the bound compounds. Then, the solution was collected and concentrated to a dry state. Subsequently, the dry residue was further dissolved with 0.2 mL methanol and subjected to HPLC analysis. As a negative control, α-glucosidase was inactivated at 100 °C in a water bath for 10 min. The α-glucosidase inhibitors in SGR2 were subsequently identified by comparing the HPLC chromatograms of active α-glucosidase and inactive α-glucosidase. The specific binding factors were calculated according to Equation (2):Specific binding factor (%) = (A_a_ − A_b_)/A_c_ × 100%(2)
where, A_a_ is the peak area of the experiment with active α-glucosidase, A_b_ is the peak area of the control group with denatured α-glucosidase, and A_c_ is the peak area of the extract.

### 3.5. Extraction and Isolation

The SGR2 extract (38 g) was inserted into an MCI gel CHP-20P column with a MeOH/H_2_O gradient (10:90 → 100:0, *v*/*v*) to yield five fractions (Fr 2.1–Fr 2.5). Guided by the HPLC analysis results, a combination of MCI macroporous resin column chromatography, HSCCC, and semi-preparative HPLC was conducted to isolate the active peaks. A total of 16 compounds were isolated: compounds **1** (25.8 mg), **2** (42.7 mg), **3** (16.1 mg), **4** (18.6 mg), **5** (30.7 mg), **6** (15.0 mg), **7** (14.6 mg), **8** (138.8 mg), **9** (14.3 mg), **11** (36.7 mg), **12** (185.5 mg), **13** (8.8 mg), **14** (10.8 mg), **15** (10.7 mg), **16** (5.0 mg), and 17 (3.8 mg). However, compound 10 could not be elucidated because it was unstable after purification. A flowchart of the process is shown in Figure 2. (Details of the extraction and isolation are presented in the Appendix A).

### 3.6. Spectroscopic Data

The optical rotations were determined using a JASCO P-2000 polarimeter (Jasco Co., Tokyo, Japan). The IR spectra were recorded on a Mattson Genesis II spectrometer (Thermo Nicolet, Madison, WI, USA). High-resolution electrospray ionization (HR) ESI-MS spectra were obtained using an LCMS-IT-TOF (Shimadzu, Tokoyo, Japan) mass spectrometer with acetonitrile as the solvent. ^1^H and ^13^C NMR spectra were recorded on a Brucker Avance 500 instrument (Bruker BioSpin, Billerica, MA, USA) (500 MHz for ^1^H NMR and 126 MHz for ^13^C NMR) with C_5_D_5_N as the solvent. UV spectra were recorded on a Shimadzu UVmini-1240 (Shimadzu, Kyoto, Japan).

#### 3.6.1. Siraitic Acid III E (**4**)

White amorphous powder; [α${]}_{D}^{20}$ +15.1 (c 0.1, MeOH); IR (KBr) ν_max_: 3442, 1698, 1649, 1074 cm^−1^; UV (ACN) λ_max_: 247 nm; ^1^H and ^13^C NMR (Table 3 and Table 4); HR-ESI-MS *m*/*z*: 987.4471 [M+HCOO]^−^ (calculated for C_47_H_71_O_22_, 987.4442).

#### 3.6.2. Siraitic Acid IIb E (**6**)

White amorphous powder; IR (KBr) ν_max_: 3391, 2927, 1704, 1647, 1076 cm^−1^; UV (ACN) λ_max_: 234 nm; ^1^H and ^13^C NMR (Table 3 and Table 4); HR-ESI-MS *m*/*z*: 803.3812 [M+Na]^+^ (calculated for C_40_H_60_O_15_Na, 803.3824).

#### 3.6.3. Siraitic Acid II E (**7**)

White amorphous powder; IR (KBr) ν_max_: 3400, 2927, 1703, 1647, 1072 cm^−1^; UV (ACN) λ_max_: 228 nm; ^1^H and ^13^C NMR (Table 3 and Table 4); HR-ESI-MS *m*/*z*: 803.3774 [M+Na]^+^ (calculated for C_40_H_60_O_15_Na, 803.3824).

#### 3.6.4. Siraitic Acid IV H (**8**)

White amorphous powder; [α${]}_{D}^{20}$ −394.8 (c 0.1, MeOH); IR (KBr) ν_max_: 3417, 3168, 1716, 1069 cm^−1^; UV (ACN) λ_max_: 231 nm; ^1^H and ^13^C NMR (Table 3 and Table 4); HR-ESI-MS *m*/*z*: 1103.5267 [M−H]^−^ (calculated for C_53_H_83_O_24_, 1103.5280).

#### 3.6.5. Siraitic Acid II G (**9**)

White amorphous powder; IR (KBr) ν_max_: 3400, 2929, 1703, 1072 cm^−1^; UV (ACN) λ_max_: 232 nm; ^1^H and ^13^C NMR (Table 3 and Table 4); HR-ESI-MS *m*/*z*: 823.4124 [M+HCOO]^−^ (calculated for C_42_H_63_O_16_, 823.4122).

#### 3.6.6. Siraitic Acid II A (**11**)

White amorphous powder; [α${]}_{D}^{20}$ +65.9 (c 0.1, MeOH); IR (KBr) ν_max_: 3391, 2933, 1698, 1069 cm^−1^; ^1^H and ^13^C NMR (Table 3 and Table 4); HR-ESI-MS *m*/*z*: 987.4471 [M+HCOO]^−^ (calculated for C_47_H_71_O_22_, 987.4442).

### 3.7. HPLC Conditions

HPLC analysis was performed on an Agilent 1260 Infinity II (Agilent Technologies, Santa Clara, CA, USA) consisting of a G7111A quaternary pump, a G7129A automated sample injector, a G7116A thermostated column compartment, a G7115A DAD detector, and a Poroshell 120 EC-C18 column (4 μm, 150 mm × 4.6 mm, Agilent Technologies, Santa Clara, CA, USA) at 30 °C. The gradient system consisted of acetonitrile (A) and water (B), using the following gradient elution profile: 0 min, 20% A, 40 min, 50% A, at a flow rate of 0.8 mL/min.

### 3.8. Determination of Sugar Configuration

The sugar configurations of compounds **4**, **6**, **7**, **8**, **9**, and **11** were determined using a method described previously [46]. Briefly, the compound (1 mg) was hydrolyzed by heating in 0.5 M HCl (0.1 mL), and Amberlite IRA400 was added to neutralize it. After the filtrate was vacuum-dried, the residue was dissolved in 0.1 mL pyridine that contained 0.5 mg *L*-cysteine methyl ester hydrochloride and heated at 60 °C for 1 h. Subsequently, a 0.1 mL solution of o-tolyl isocyanate (0.5 mg) in pyridine was added, and the mixture was heated at 60 °C for another 1 h. The reaction solution was directly analyzed by HPLC on a Poroshell 120 EC-C18 column (4 μm, 150 mm × 4.6 mm); an isocratic elution of acetonitrile and water (25:75, *v*/*v*) at a flow rate of 0.8 mL/min was performed, and the detection wavelength was 254 nm. The absolute configuration of the sugar component was determined by comparing the retention time t_R_ for these compounds with that of *D*-glucose.

### 3.9. Molecular Docking Analysis

To explore how the screened inhibitors conjugated with α-glucosidase, the binding interactions between compounds and α-glucosidase were investigated using molecular docking. The crystal structure of α-glucosidase was downloaded from the Protein Data Bank database (PDB ID: 3A4A) (https://www.rcsb.org/, accessed on 16 February 2023), and the 3D structures of the ligands with the lowest energies were established using Chem3D Ultra 19.0. Molecular docking was performed using the AutoDock vina 1.1.2 software suite [47]. The centroid coordinate of the grid box was set as below: x: 21.281, y: −0.635, z: 18.475, with a map of 33.75 × 33.75 × 33.75.

### 3.10. Statistical Analysis

All experiments were repeated three times, and the results are presented as the mean ± standard deviation. The data were analyzed using the SPSS 22 and Origin 2018 software. (OriginPro 2018C SR1 b9.5.1.195).

## 4. Conclusions

Elucidating the phytochemical profiles and bioactive compounds is crucial for the application of new plant resources in the health, food, and pharmaceutical industries. In the present study, a strategy for rapid screening and targeted separation of α-glucosidase inhibitors from an active *S. grosvenorii* roots fraction, SGR2, was developed using UF-HPLC and a combination of comprehensive separation techniques. Sixteen potential inhibitors were successfully isolated from SGR2, including lignans, cucurbitacins, and cucurbitane glycosides. Among them, six compounds were novel cucurbitane-type triterpenoids, and their structures were meticulously elucidated through NMR spectroscopy and HR-ESI-MS. Notably, all isolated compounds exhibited α-glucosidase inhibitory activities, as verified by enzyme inhibition assays and molecular docking analysis. Siraitic acid II C (compound **14**) showed the highest inhibitory activity with an IC_50_ value of 430.13 ± 13.33 μM, which is superior to that of acarbose (1332.50 ± 58.53 μM). Molecular docking evaluations confirmed that these inhibitors interacted with α-glucosidase through hydrogen bonds and hydrophobic interactions, showing different active mechanisms against α-glucosidase than that exhibited by acarbose. Therefore, our results not only demonstrate the beneficial effects of *S. grosvenorii* roots and their constituents on α-glucosidase inhibition, providing a theoretical basis for their application in the prevention and treatment of diabetes, but also indicate that this strategy is efficient in distinguishing potential α-glucosidase inhibitors from natural plant extracts, given that it overcomes the randomness and blindness associated with the isolation of compounds using existing conventional separation strategies.

## Figures and Tables

**Figure 1 ijms-24-10178-f001:**
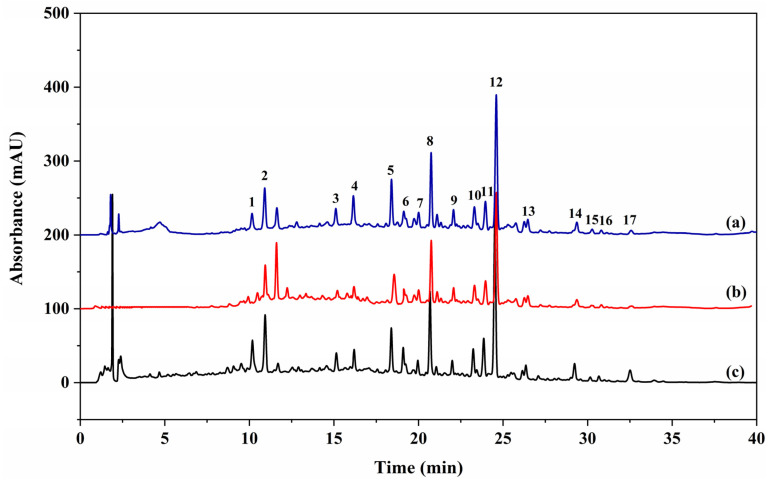
The chromatograms obtained from the high-performance liquid chromatography (HPLC) (230 nm) of the chemical constituents in the SGR2 fraction obtained by ultrafiltration. The black solid line represents HPLC profiles of the SGR2 fraction without ultrafiltration (c); the blue and red lines represent HPLC profiles of the SGR2 fraction with activated (a) and inactivated α-glucosidase by ultrafiltration (b), respectively.

**Figure 2 ijms-24-10178-f002:**
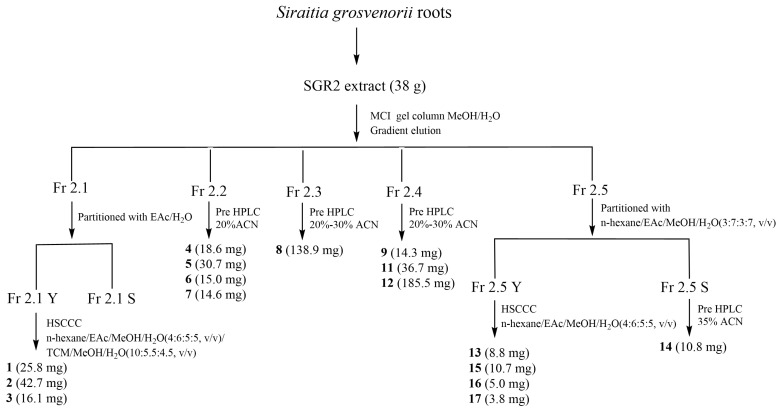
Flowchart of the process used for the purification of compounds from *Siraitia grosvenorii* roots.

**Figure 3 ijms-24-10178-f003:**
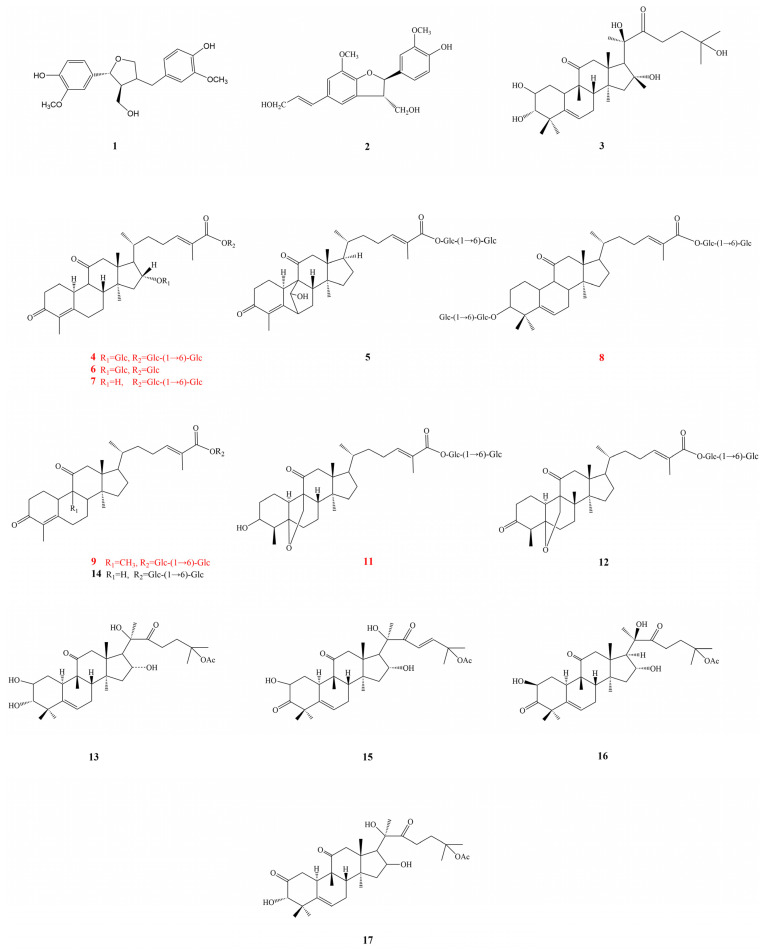
Chemical structures of compounds **1**–**17**, isolated from *Siraitia grosvenorii* root (SGR2).

**Figure 4 ijms-24-10178-f004:**
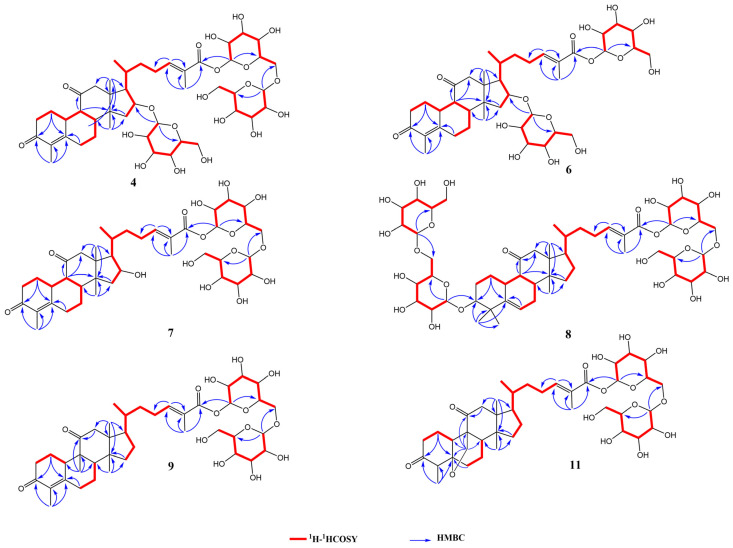
Key ^1^H ^1^H correlation spectroscopy (COSY; red bold lines) and heteronuclear multiple bond coherence (HMBC; blue arrows) correlations of compounds **4**, **6**, **7**, **8**, **9**, and **11**.

**Figure 5 ijms-24-10178-f005:**
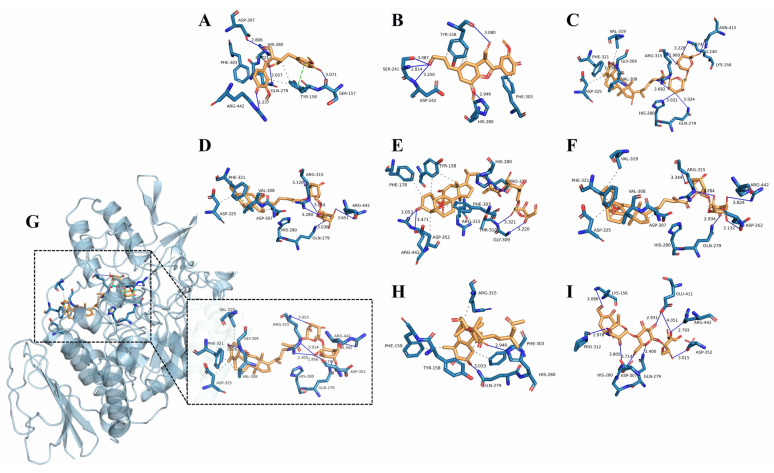
Molecular docking analysis of the isolated potential inhibitors and acarbose with α-glucosidase. Compound **1** (**A**), **2** (**B**), **7** (**C**), **9** (**D**), **11** (**E**), **12** (**F**), **14** (**G**), **15** (**H**), and acarbose (**I**). The blue solid lines represent hydrogen bonds, and the gray dashed lines represent hydrophobic interactions. Compound **14** was used as an example to demonstrate the three-dimensional structure and pocket of the protein-ligand complex.

**Table 1 ijms-24-10178-t001:** Inhibition of α-glucosidase by the crude *S. grosvenorii* root extract and its MeOH eluent fractions.

Sample	Inhibition (%) ^1^	IC_50_ (μg/mL) ^2^
crude extract	9.81 ± 1.21	4850.0 ± 217.6
SGR1	48.09 ± 0.83	990.7 ± 21.6
SGR2	59.07 ± 2.73	631.1 ± 50.2
Acarbose	55.15 ± 1.55	860.3 ± 31.3 ^3^

^1^ The results are expressed as % of inhibition testing the samples at a concentration of 1.0 mg/mL; ^2^ each value is mean ± SD; ^3^ the molar concentration is 1332.50 ± 58.53 μM.

**Table 2 ijms-24-10178-t002:** The retention time, specific binding factors, and inhibitory activity of 17 active peaks binding to α-glucosidase (%).

Peak	Retention Time (min)	Compound Names	Specific Binding Factors Mean ± SD (n = 3)	Inhibition (%) ^1^	IC_50_ (μM) ^2^
**1**	10.16	(-)-lariciresinol	4.97 ± 0.15	78.18 ± 1.54	1832.87 ± 31.33
**2**	10.91	3,4′-dimethoxy-4,9,9′-trihydroxy-benzofuranolignan-7′-ene	1.87 ± 0.77	63.43 ± 0.23	2275.47 ± 13.80
**3**	15.12	23,24-dihydrocucurbitacin F	3.76 ± 0.33	38.27 ± 0.70	n.d.
**4**	16.19	Siraitic acid III E	2.93 ± 0.72	36.74 ± 0.36	n.d.
**5**	18.40	Siraitic glycoside II F	4.39 ± 0.42	42.73 ± 1.79	n.d.
**6**	19.13	Siraitic acid IIb E	3.91 ± 0.17	42.44 ± 0.81	n.d.
**7**	20.00	Siraitic acid II E	3.46 ± 1.03	58.48 ± 1.19	1206.84 ± 5.49
**8**	20.74	Siraitic acid IV H	0.27 ± 0.18	23.38 ± 1.05	n.d.
**9**	22.07	Siraitic acid II G	5.33 ± 2.11	53.12 ± 0.99	1580.96 ± 12.54
**10**	23.30	Unknown	6.27 ± 1.08	-	-
**11**	23.96	Siraitic acid II A	6.59 ± 0.91	50.60 ± 1.23	1239.78 ± 20.49
**12**	24.60	Siraitic acid II B	6.65 ± 1.18	51.34 ± 0.83	1034.53 ± 36.95
**13**	26.46	23,24-dihydrocucurbitacin F-25-acetate	1.31 ± 0.06	21.67 ± 0.58	n.d.
**14**	29.36	Siraitic acid II C	17.26 ± 3.14	72.01 ± 2.18	430.13 ± 13.33
**15**	30.27	Cucurbitacin B	7.98 ± 1.11	50.59 ± 0.98	1505.41 ± 57.02
**16**	30.81	23,24-dihydrocucurbitacin B	4.70 ± 0.76	27.54 ± 3.80	n.d.
**17**	32.58	Dihydroisocucurbitacin B-25-acetate	2.28 ± 0.44	23.74 ± 0.33	n.d.

^1^ Inhibition rate (%) was measured in the concentration of 1.0 mg/mL compound; ^2^ n.d. = not determined because α-glucosidase inhibition at a concentration of 1.0 mg/mL was lower than 50%.

**Table 3 ijms-24-10178-t003:** ^1^H-NMR data of compounds **4**, **6**–**9**, and **11** (in C_5_D_5_N, *δ* in ppm, 500 MHz).

Position	4	6	7	8	9	11
**aglycone**						
**1**	2.60, 1.42, m	2.59, 1.41, m	2.65, 1.51, m	1.85, m	2.02, 1.87, m	1.11, 2.05, m
**2**	2.47, 2.38, m	2.47, 2.38, m	2.53, 2.44, m	2.58, 2.16, m		1.54, 1.91, m
**3**				3.73, m		3.86, d (7.5)
**4**						1.58, dd (7.1, 3.0)
**6**	2.79, 1.80, m	2.80, 1.78, m	2.84, 1.86, m	5.51, m	2.45, m	1.84, 1.98, m
**7**	1.51, 1.24, m	1.48, 1.23, m	1.58, 1.33, m	1.59, 1.19, m	1.68, m	1.25, 1.96, m
**8**	1.82, m	1.82, m	1.86, m	1.96, m	1.90, m	2.04, br t (7.2)
**9**	2.27, m	2.28, t, (11.1)	2.37, t (11.4)	2.18, m		
**10**	2.72, m	2.71, m	2.75, m	3.14, m	2.91, m	2.40, dd (11.8, 5.8)
**11**						
**12**	2.80, 2.47, m	2.80, 2.47, m	2.93, 2.56, d (12.4)	2.50, m	2.82, 2.61, d (16.3)	
**15**	1.64, 1.95, m	1.93, 1.62, m	1.95, 1.68, m	1.17, m	1.29, m	
**16**	4.52, m	4.52, t (7.0)	4.30, m	1.89, m	1.94, m	
**17**	2.21, m	2.13, m	2.12, m		1.70, m	
**18**	0.75, s	0.74, s	0.83, s	0.70, s	0.76, s	0.68, s
**19**					1.11, s	3.67, d (8.4), 4.73, d (8.4)
**20**	1.60, m	1.61, m	1.61, m	1.30, m	1.34, m	
**21**	0.86, d (6.0)	0.87, d (6.6)	0.93, d (6.2)	0.82, d (6.4)	0.84, d (6.5)	0.85, d (6.4)
**22**	2.11, 1.72, m	2.07, 1.79, m	1.98, m	1.37, 0.98, m	1.44, m	
**23**	2.36, 2.08, m	2.44, 2,10, m	2.47, 2.04, m	2.15, 1.94, m	2.19, m	
**24**	7.36, t (7.7)	7.40, t (7.1)	7.38, t (6.7)	7.06, t (6.9)	7.09, t (7.5)	
**26**	1.93, s	1.95, s	1.93, s	1.91, s	1.92, s	1.93, s
**28**	1.91, s	1.91, s	1.89, s	1.11, s	1.90, s	1.35, d (7.0)
**29**				1.52, s		
**30**	1.50, s	1.51, s	1.48, s	0.90, s	1.16, s	1.19, s
sugar						
G_I_1	4.87, d (7.7)	4.88, d (7.7)	6.42, d (8.2)	4.81, m	6.45, d (8.0)	6.47, d (8.0)
G_I_2	4.01, m	3.95, t (8.5)	4.16, m	3.90, m	4.25, m	4.01, m
G_I_3	4.24, m	4.23, m	4.15, m	4.20, m	3.89, m	3.90, m
G_I_4	4.16, m	4.17, m	4.41, t (9.7)	3.97, m	4.44, m	4.21, m
G_I_5	3.92, m	4.01, m	4.28, m	4.11, m	4.22, m	4.18, m
G_I_6	4.34, 4.46, m	4.44, 4.37, m	4.77, 4.37, m	4.83, 4.30, m	4.79, d (11.9) 4.38, m	4.78, 4.38, m
G_II_1	6.37, d (7.6)	6.50, d (7.6)	6.47, d (7.6)	5.15, m	5.02, m	4.93, d (8.3)
G_II_2	4.25, m	4.34, m	4.01, t (8.2)	4.04, m	4.05, t (8.2)	4.01, m
G_II_3	4.19, m	4.31, m	3.90, m	4.20, m	4.31, m	4.01, m
G_II_4	4.20, m	4.36, m	4.21, m	4.11, m	4.24, m	4.18, m
G_II_5	4.24, m	4.02, m	4.18, m	4.20, m	4.22, m	3.85, m
G_II_6	4.73, 4.34, m	4.66, 4.42, m	4.50, 4.36, m	4.53, 4.37, m	4.49, d (11.9) 4.37, m	4.63, 4.40, m
G_III_1	4.99, d (7.9)			6.43, d (7.6)		
G_III_2	4.01, m			4.22, m		
G_III_3	4.01, m			4.23, m		
G_III_4	4.20, m			4.41, m		
G_III_5	3.87, m			4.20, m		
G_III_6	4.65, 4.42, m			4.77, 4.37, m		
G_IV_1				5.02, d (7.8)		
G_IV_2				4.01, m		
G_IV_3				3.88, m		
G_IV_4				4.23, m		
G_IV_5				4.20, m		
G_IV_6				4.47, 4.35, m		

**Table 4 ijms-24-10178-t004:** ^13^C-NMR data for compounds **4**, **6**–**9**, and **11** in C_5_D_5_N (δ in ppm, 125 MHz).

Position	4	6	7	8	9	11	Position	4	6	7	8	9	11
**aglycone**							sugar						
**1**	28.8	28.8	28.9	26.2	24.9	18.5	G_I_1	102.8	102.8	96.6	107.5	96.6	96.7
**2**	37.6	37.6	37.7	30.2	37.8	20.2	G_I_2	75.4	75.5	74.5	75.7	74.5	75.6
**3**	198.9	198.8	198.9	88.2	197.9	71.0	G_I_3	79.0	78.7	78.2	79.0	78.9	78.3
**4**	130.3	130.3	130.3	43.0	132.9	40.8	G_I_4	72.6	72.8	71.1	72.1	71.2	71.3
**5**	158.2	158.1	158.2	144.8	157.3	86.4	G_I_5	78.7	79.0	78.9	77.8	78.3	79.0
**6**	31.2	31.2	31.3	118.9	28.5	26.2	G_I_6	62.9	62.4	69.8	70.9	69.9	70.0
**7**	27.9	27.9	27.9	28.7	21.7	28.5	G_II_1	96.5	96.5	105.8	106.0	105.7	105.8
**8**	45.3	45.3	45.4	36.5	45.8	46.0	G_II_2	74.4	74.7	75.5	75.6	75.5	74.6
**9**	55.0	55.0	55.1	50.7	50.0	60.6	G_II_3	78.6	79.0	78.7	79.0	78.8	78.9
**10**	38.1	38.1	38.1	35.4	41.1	45.4	G_II_4	71.8	71.3	71.8	72.1	71.8	71.9
**11**	210.9	210.8	211.0	212.5	214.8	210.3	G_II_5	78.8	79.7	78.8	78.9	78.8	78.8
**12**	52.5	52.6	52.8	52.3	51.5	51.0	G_II_6	69.8	64.2	62.9	63.2	63.0	63.0
**13**	47.7	47.8	47.9	47.0	47.4	49.3	G_III_1	105.5			96.7		
**14**	51.1	51.1	51.5	47.8	49.7	49.6	G_III_2	75.4			74.6		
**15**	40.0	40.0	45.4	33.4	34.9	34.2	G_III_3	78.8			78.9		
**16**	83.0	82.8	77.0	57.8	28.3	32.6	G_III_4	73.5			71.2		
**17**	56.3	56.3	58.7	50.5	50.8	49.9	G_III_5	78.8			78.4		
**18**	17.9	17.9	17.9	17.1	17.9	17.0	G_III_6	64.0			70.0		
**19**					23.2	74.9	G_IV_1				105.8		
**20**	34.9	34.8	35.2	36.7	36.5	36.4	G_IV_2				75.6		
**21**	19.2	19.2	19.1	18.8	18.6	18.7	G_IV_3				78.8		
**22**	33.8	33.9	34.9	35.5	35.2	35.3	G_IV_4				71.8		
**23**	26.2	26.0	26.2	26.3	26.2	26.2	G_IV_5				78.7		
**24**	146.4	146.3	145.5	145.2	145.0	144.9	G_IV_6				63.0		
**25**	127.2	127.3	127.8	128.0	127.9	127.9							
**26**	13.0	13.1	12.9	13.0	12.8	12.9							
**27**	167.6	167.6	167.5	167.6	167.5	167.6							
**28**	11.5	11.6	11.6	27.3	11.6	13.5							
**29**				25.6									
**30**	17.7	17.7	17.9	17.2	19.7	20.4							

**Table 5 ijms-24-10178-t005:** Molecular interactions between α-glucosidase and potential inhibitors/acarbose.

Main Compound	Binding Affinity (kcal/mol)	Number of Binding to Residues	Residues Involved in H-Bond Formation	Hydrophobic Interaction
**1**	−8.5	5	HIS 280, GLN 279, SER 157, ASP 307, ARG 442	SER 157, GLN 279, HIS 280, ASP 307, ARG 442 π-π stacking: TYR158
**2**	−8.5	5	SER 241, ASP 242, TYR 158, HIS 280	TYR 158, SER 241, ASP 242, HIS 280
**7**	−9.9	8	LYS 156, HIS 280, GLN 279, ARG 315, GLY 309, ASN 415	VAL 319, VAL 308, ASP325, PHE 321
**9**	−9.9	5	ARG 315, ARG 442, GLN 279	VAL 308, PHE 321, ASP 325, ASP 307
**11**	−9.8	5	HIS 280, THR 310, GLY 309, ASP 352, ARG 442	ASP 352, PHE 303, ARG 315, TYR 158, PRO 312
**12**	−10.0	5	GLN 279, ARG 315, ASP 352, ARG442	VAL 319, PHE 321, ASP325, ASP 307
**14**	−10.0	7	ARG 315, ASP 352, GLN 279, GLY 309, ARG442	VAL 319, PHE 321, ASP 325, VAL 308
**15**	−8.4	2	GLN 279, HIS 280	PHE159, PHE 303, ARG 315, TYR 158
**Acarbose**	−8.4	9	GLN 279, ASP352, ARG 442, GLU 411, PRO 312, HIS 280, LYS 156	-

## Data Availability

The data presented in this study are available upon request from the corresponding author.

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
