# Peer review of "Identification and Isolation of α-Glucosidase Inhibitors from Siraitia grosvenorii Roots Using Bio-Affinity Ultrafiltration and Comprehensive Chromatography"

_ijms, 2023, doi:10.3390/ijms241210178_

Round 1

Reviewer 1 Report

The manuscript presented by Lu et al. describes the identification and isolation of different triterpenoic kind alpha-glucosidase inhibitors by combining bio-affinity ultrafiltration and chromatographic methods.

The manuscript is concisely and intelligibly written and quite good to read. The references are, as far as I can tell, sufficient and most of the applied techniques are described elaborately. A main part of the manuscript deals with the identification of the isolated compounds by nmr spectroscopy. Therefore, nmr spectroscopy should be adressed in the materials and methods section as a method (which spectrometer has been used, which parameters have been used and so on) just like the other methods too (HPLC for example is described in detail...). Maybe I have missed it, but the authors also used IR and UV/Vis spectroscopy? Same question here, which machines have been used, sample preparation etc.

Literature known compounds have been cited and nmr spectroscopic data have been added to the supporting information, but it would be useful, if also the spectra are shown. Some of the cited research articles concerning these data are 30-40 years old and do not show any spectra but only give evaluation of the, in that time, recent data. In special: compound 2 should be described in citation 32 but if I have seen it correctly the aldehyde is described and not the primary alcohol (please check again).

Nonetheless, a lot of hard work has been put into this research, especially the evaluation of the nmr spectra and identification of the isolated compounds. Here only one question: In Figure 4, compound 11 you indicate heteronuclear coupling between the proton on C8 and the methyl group on C14. Are the authors sure, they see a coupling between these since they show trans konfiguration. Most of the compounds have been isolated in amounts where elemental analysis could have been possible. Though the analytical data are in terms of modern standards sufficient an additional prove of purity by elemental analysis would have been good, in special in terms of the inhibitor screening.

Concerning the screening of the potential inhibitors: It's obvious to use mass concentrations for the crude and the processed extracts SGR1 and SGR2, but in case of the isolated compounds inhibition should also be specified in molar concentrations for easier comparison to other publications. In table 2 it is stated, that for various compounds IC50 values haven't been determined, because the inhibition at a concentration of 1 mg/ml was below 50%. But, if I see it correctly, compounds 9, 12 and 15 have IC50 values above 1 mg/ml indicating, that these also should have inhibitions at 1mg/ml below 50%. The inhibition rate shown at 1 mg/ml nonetheless is for all 3 compounds above 50%. Please explain.

Concerning the molecular docking analysis: I'm no expert here and therefore only can state, that the analysis/simulation seems coherent to me.

Overall, after adding/dealing with the mentioned minor revisions, I can recommend publishing this manuscript in IJMS.

Author Response

We feel great thanks for your professional review work on our article. As you are concerned, there are several problems that need to be addressed. According to your nice suggestions, we have made extensive corrections to our previous draft, the detailed corrections are listed below.

1. Thank you for your careful examination. We are sorry for our carelessness. We have added NMR, IR and UV spectroscopy and optical rotationsin the materials and methods section as a method. We carefully checked the reference lists and corrected the erroneous citations of compounds 1 and 2.

2. It is confirmed that there is heteronuclear coupling between the proton on C-8 and the methyl on C-14 in HMBC spectrum, and we have redrawed the Figure 4. However, we believe the NMR, HR-ESI-MS and HPLC analysis are sufficient to prove the purity of the isolated compounds, we are sorry for having not do elemental analysis.

3. The IC50 values of compounds 9 and 12 have above 1 mg / mL, which may be caused by errors in the fitting process. The IC50 value of compound 15 was recalculated to be IC50 = 840.50±31.83ug / mL, and we are sorry for our carelessness. Meanwhile, we believe that it is a good suggestion to designate the inhibitory concentration of the isolated compound as the molar concentration, so we modified it in the revised draft. 

Reviewer 2 Report

In the manuscript titled “Identification and isolation of α-glucosidase inhibitors from Siraitia grosvenorii roots using bio-affinity ultrafiltration and comprehensive chromatography”, Lu and co-workers report their investigation on the antidiabetic activities of 17 new potential α-glucosidase inhibitors extracted from S. grosvenorii roots by means of efficient an UF-HPLC separation method.  In general, the manuscript is well organized, the methods used are appropriate, the results are clearly presented and discussed, and accurately compared with results already published in the literature in the field.

However, the following point should be addressed before recommending publication in IJMS:

- The "Methods" section does not provide any info about the definition of the docking grid generation, eventual validation of the docking experiment (RMSD of the co-crystalized ligand), and the parameters used for protein and ligand optimization. It is suggested to improve this section.

- It should be interesting to report an image of the best compound in complex with the target protein, in order to appreciate also the tridimensional orientation of the molecule into the binding pocket.

Minor editing of English language required

Author Response

We feel great thanks for your professional review work on our article. According to your nice suggestions, we have made extensive corrections to our previous draft, the detailed corrections are listed below.

1.As suggested by the reviewer, we have added docking grid box spacing that was constructed around the binding site and coordinates generation.

2.We added the three-dimensional orientation of the molecule of compound 14 into the binding pocket in Figure 5.